# GAUSSUNVEIL: UNIFIED OCCLUSION-AWARE GAUSSIAN REFINEMENT FOR SPARSE-VIEW SCENE RECONSTRUCTION

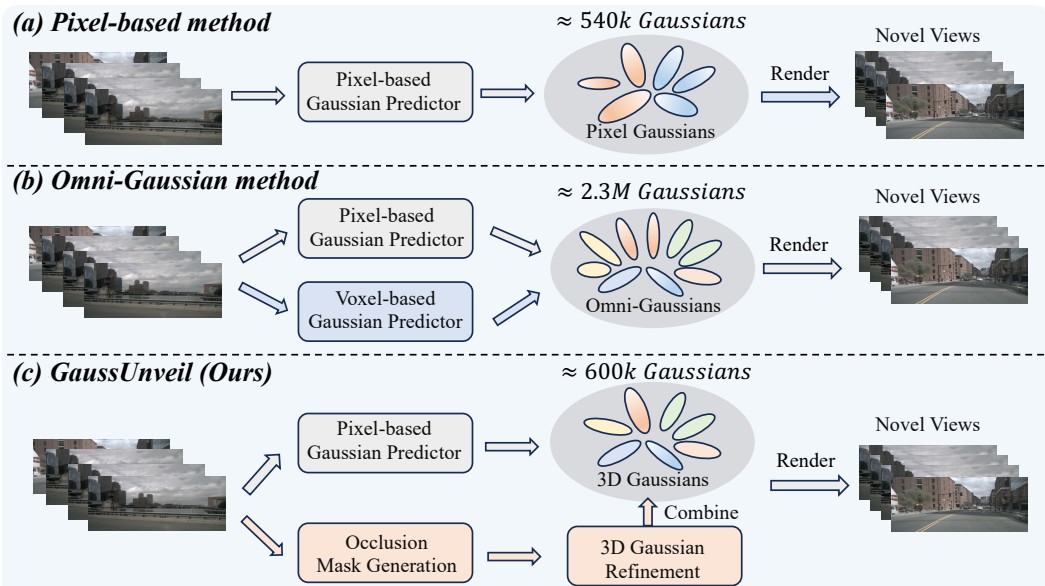

Figure 1: Comparison of different Gaussian-based reconstruction pipelines. (a) Pixel-based methods reconstruct the multi-view scenes by predicting per-pixel Gaussians, resulting in ≈540K Gaussians. (b) The Omni-Gaussian method employs both pixel- and voxel-based predictors, thereby improving reconstruction quality while introducing significant redundancy (≈2.3M Gaussians). (c) Our GaussUnveil selectively refines only occlusion-prone regions, achieving comparable quality with far fewer Gaussians (≈600K).

## ABSTRACT

Ego-centric 3D reconstruction from sparse, low-overlap views is challenging, as cross-view correspondences are limited, occlusions occur frequently, and per-camera frusta often truncate scene structures. Explicit Gaussian pipelines mitigate some of these challenges, and the dual-branch methods that couple pixel- and volume-based Gaussians (e.g., Omni-Scene) further enhance robustness. However, they typically refine large numbers of Gaussians uniformly, regardless of visibility or structural ambiguity. We propose GaussUnveil, an occlusion-aware selective-refinement framework that shifts the paradigm from *refining everywhere* to *refining where it matters*. By *unveiling* regions of uncertainty near occlusions, GaussUnveil identifies where additional Gaussian refinement is needed. Specifically, we derive occlusion masks from depth-gradient discontinuities, lift them into the 3D volume to initialize a compact set of Gaussian queries. Then, we employ a lightweight refinement block that aggregates self-context and multi-view features while iteratively updating the mean and covariance of each Gaussian query under differentiable rendering. Extensive experiments on both ego-centric and scene-centric benchmarks demonstrate the effectiveness of the proposed method compared to the state-of-the-art reconstruction methods. For instance, GaussUnveil delivers superior performance while using about 30% fewer Gaussians and is approximately 34% faster than Omni-Scene.

# 1 INTRODUCTION

*"I cannot see the true face of Mount Lu, for I am within this very mountain."*

—— Su Shi, Inscription on the Wall of Xilin Temple

Sparse-view scene reconstruction aims to recover 3D structures from only a few input views, and has become a fundamental problem in computer vision, contributing to various downstream tasks in autonomous driving Wang et al. (2024); Tang et al. (2024); Huang et al. (2021); Li et al. (2022b;a); Liu et al. (2023); Hu et al. (2023); Jiang et al. (2023); Jia et al. (2023). Recent advances (Yu et al., 2021b; Wang et al., 2021; Liu et al., 2022; Chen et al., 2021; Johari et al., 2022) have incorporated 3D structural priors into neural networks, enabling the prediction of implicit neural fields (Mildenhall et al., 2020), light fields (Suhail et al., 2022), or explicit 3D Gaussians (Kerbl et al., 2023) in a single forward pass. Among them, Gaussian-based methods have shown clear advantages in both inference speed and visual quality, benefiting from their explicit parameterization and the efficiency of rasterization-based differentiable rendering.

A central design choice in Gaussian-based methods lies in how Gaussians are parameterized. Pixel-based Gaussians (Chen et al., 2024; Charatan et al., 2024) predict per-pixel depths and unproject them into 3D along camera rays, producing detailed reconstructions when dense overlaps exist. However, these methods rely on substantial cross-view overlap, an assumption that seldom holds in practice, especially for autonomous driving. In ego-centric settings, the overlaps are small and objects are often occluded or truncated, which introduces scale ambiguity and leads to frequent failures. Volume-based Gaussians (Huang et al., 2024; Zuo et al., 2025), in contrast, directly lift features into 3D space, where volumetric continuity allows partial completion of occluded or truncated regions. This makes them more robust under sparse observations, but their bounded extent prevents recovery of distant structures, and their cubic complexity constrains resolution, leading to missing fine-grained details. Notably, Omni-Scene (Wei et al., 2025) fuses pixel- and volume-based Gaussians to exploit complementary cues and achieves strong performance.

However, this dual-branch architecture introduces substantial redundancy with large numbers of Gaussians regardless of visibility or geometric certainty, including well-observed regions and occluded areas. Further analysis reveals that the voxel-based branch places Gaussians in every voxel of the 3D grid, often reaching millions of Gaussians, far more than the pixel-based branch. As stated in Wei et al. (2025), most regions are sufficiently observed and can be accurately reconstructed by pixel Gaussians alone, while voxel Gaussians primarily contribute near occlusions and other visibility gaps. This observation motivates us to present a unified pipeline that produces a coarse reconstruction, then identifies occluded or uncertain regions and restricts Gaussian refinement to those regions only. By reframing the task from *refine everywhere* to *refine where it matters*, we significantly cut redundant Gaussians and computational overhead as shown in Figure 1.

In this paper, we propose GaussUnveil, a lightweight yet effective framework that predicts pixel-based Gaussians from multi-view inputs and performs 3D refinement only to regions likely affected by occlusion. *Our key insight is that uncertainty in ego-centric scenes concentrates at visibility transitions, so we localize unreliable geometry and refine only the affected Gaussians to preserve accuracy while reducing redundant computation.* To be specific, we interpret sharp depth-gradient changes as visibility boundaries and convert them into a narrow-band uncertainty region via thresholding and kernel dilation. The resulting occlusion mask serves as a low-cost, robust *where-to-refine* prior that localizes likely occlusions and geometric discontinuities. We further introduce a lightweight Refine Block that targets uncertain regions by initializing a set of queries to instantiate 3D Gaussians and updating them via interleaved self-aggregation, cross-view aggregation, and Gaussian refinement layers.

GaussUnveil exhibits properties absent from prior models: (1) it identifies likely occluded regions across different views with a simple forward pass; (2) by restricting Gaussian refinement updates to the *where-to-refine* regions, it dramatically reduces the number of Gaussians that must be rendered. We evaluate the effectiveness of GaussUnveil on both ego-centric and scene-centric benchmarks and show promising results compared with the state-of-the-art methods. Notably, GaussUnveil reduces the number of Gaussians by up to 30% while still exhibiting performance gains in reconstruction quality on nuScenes. Our contributions are summarized as follows:

- We propose GaussUnveil, a unified 3D Gaussian framework that predicts pixel-based Gaussians and selectively refines regions likely affected by occlusion.

- We introduce a compact 3D refinement block that iteratively updates Gaussians only in masked regions, enabling recovery of fine-grained details by refined Gaussians;

- Extensive experiments on several reconstruction benchmarks demonstrate that GaussUnveil reduces the Gaussians by up to 30% while achieving state-of-the-art performance.

## 2 RELATED WORK

**3D Scene Reconstruction.** Neural radiance fields model a scene as a continuous volumetric function and optimize it by backpropagation. NeRF achieves high-fidelity novel views but needs dense per-ray sampling, so even accelerated variants still carry notable computational cost and often require per-scene optimization with dense captures (Mildenhall et al., 2020; Yu et al., 2021a; Müller et al., 2022; Johari et al., 2022; Barron et al., 2021; Tancik et al., 2022). To avoid per-scene training, feed-forward implicit methods inject 3D priors into the network. NeRF-based pipelines estimate radiance fields using multi-view attention or projective cues such as epipolar geometry and cost volumes, yet they inherit the expensive ray querying and remain slow at both training and inference (Yu et al., 2021b; Wang et al., 2021; Chen et al., 2021). Light-field approaches predict per-ray colors directly from images, which improves efficiency but loses 3D interpretability and cannot recover geometry (Mildenhall et al., 2019; Sitzmann et al., 2021). Explicit Gaussian representations replace volumetric integration with rasterization. 3D Gaussian Splatting (Kerbl et al., 2023) models scenes with anisotropic Gaussians and supports real-time rendering with competitive quality. Building on this idea, recent feed-forward pipelines (Chen et al., 2021; Charatan et al., 2024) predict pixel-based Gaussians from few views while using priors such as epipolar lines, cost volumes, or multi-view attention to guide geometry. These designs are effective when cross-view overlap is large, but they degrade under occlusion and frustum truncation in scene-centric scenarios, particularly in autonomous driving applications. In this paper, we focus on sparse-view reconstruction and propose GaussUnveil to address the above limitations in ego-centric scenarios.

**Gaussian Splatting in Autonomous Driving.** Recently, there has been an explosion of research adapting 3DGS (Kerbl et al., 2023) to autonomous driving, especially for driving scene reconstruction and perception tasks (Zhou et al., 2024; Lu et al., 2024; Song et al., 2025; Huang et al., 2024; Yan et al., 2024). GaussianFormer (Huang et al., 2024) encodes scenes with semantic Gaussians, where each Gaussian acts as a flexible region of interest that carries geometric and semantic features. Per-scene reconstruction methods excel in fidelity by leveraging all available sensor data for that scene. For instance, StreetGaussians (Yan et al., 2024) model a dynamic urban street with 3DGS first, which represents the static background and moving vehicles as separate Gaussian sets and introduces a layered optimization to handle dynamic cars. In parallel, researchers have developed generalizable 3DGS models (Chen et al., 2024; Charatan et al., 2024) that can reconstruct new scenes without per-scene training, using learned priors. These are typically feed-forward networks that take a small set of images (even a single view) of a scene and directly predict a 3D Gaussian scene representation. ADGaussian (Song et al., 2025) proposes a generalizable Gaussian splatting framework designed for street view reconstruction from minimal inputs. Despite these advances, egocentric driving presents limited cross-view overlap and frequent occlusion or truncation, which makes sparse-view reconstruction particularly challenging. OmniScene (Wei et al., 2025) introduces Omni-Gaussian representation that can reach the best of both pixel and volume-based Gaussian representations for ego-centric sparse-view scene reconstruction. Although this dual-branch architecture performs well in sparse-view reconstruction, it instantiates a large number of Gaussians across 3D space, which incurs substantial computational overhead. GaussUnveil tackles this challenge by shifting the paradigm from *refining everywhere* to *refining where it matters*, preserving accuracy while substantially reducing the number of Gaussians.

## 3 PRELIMINARIES

We briefly review 3D Gaussian Splatting (3DGS) as the basis of our method. A 3D scene is represented by a finite set of Gaussians $\mathcal{G} = \{\mathcal{G}_k\}_{k=1}^N$. Each Gaussian projects to an elliptical footprint

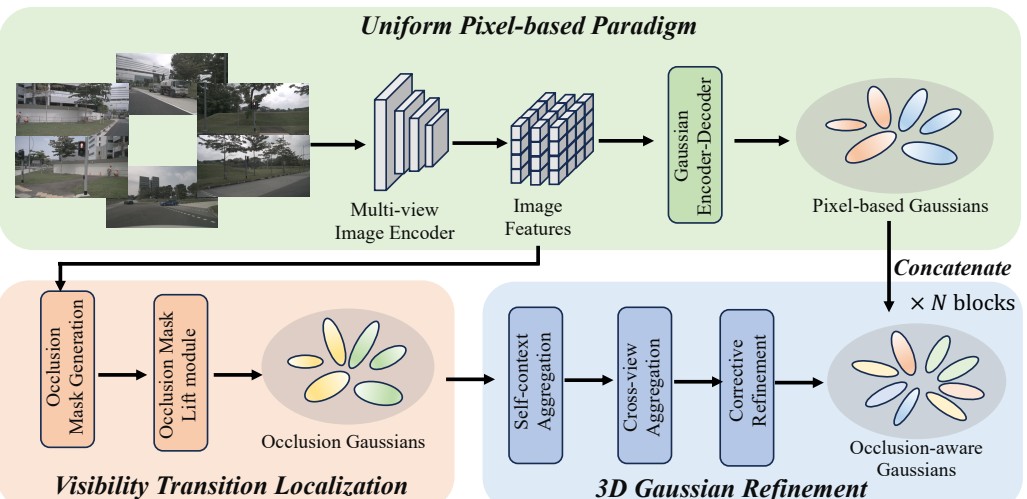

Figure 2: Overview of GaussUnveil. A uniform pixel-based pipeline (top) encodes multi-view images and decodes them into initial pixel Gaussians. To resolve visibility ambiguity, GaussUnveil localizes visibility transitions (bottom left) by deriving occlusion masks from depth gradients and lifting them into 3D to seed occlusion Gaussians. A compact 3D Gaussian refinement block (bottom right) then updates these Gaussians by the stack of self-context aggregation, cross-view aggregation, and corrective refinement layers. Finally, we concatenate the refined, occlusion-aware Gaussians with the pixel Gaussians to produce a more accurate and efficient reconstruction.

on the image plane, and the pixel color along a ray $\mathbf{r}$ is rendered by alpha compositing

$$\hat{\mathbf{I}}(\mathbf{r}) \;=\; \sum_{k=1}^{N} T_k\,\alpha_k\,\mathbf{c}_k, \qquad T_k \;=\; \prod_{j<k}\bigl(1-\alpha_j\bigr), \tag{1}$$

where $\mathbf{c}_k \in \mathbb{R}^3$ and $\alpha_k \in [0,1]$ denote the color and the opacity, respectively. This differentiable formulation allows end-to-end optimization against ground-truth images.

Pixel-based Gaussians unproject per-pixel depths into 3D and yield detailed reconstructions when view overlap is high, but they fail under occlusions and frustum truncation. Voxel-based Gaussians lift features into a 3D grid, offering volumetric continuity at the cost of cubic complexity and redundancy. Dual-branch designs such as Omni-Scene (Wei et al., 2025) combine both, but often saturate the scene with millions of Gaussians, many unnecessary in well-observed regions. In sparse-view, ego-centric settings (e.g., autonomous driving), these issues are amplified: overlaps are limited, occlusions frequent, and redundant Gaussians dominate memory and rendering cost. To formalize, we define the reconstruction objective

$$\mathcal{L}(\mathcal{G}) \;=\; \mathbb{E}_{\mathbf{r}\sim\mathcal{D}}\Bigl[\ell\bigl(\hat{\mathbf{I}}(\mathbf{r};\mathcal{G}),\,\mathbf{I}^{\star}(\mathbf{r})\bigr)\Bigr], \tag{2}$$

where $\mathbf{r}$ is a camera ray sampled from $\mathcal{D}$, $\hat{\mathbf{I}}(\mathbf{r};\mathcal{G})$ is the rendered color, $\mathbf{I}^{\star}(\mathbf{r})$ is the ground-truth color, and $\ell(\cdot,\cdot)$ is a per-ray discrepancy. We partition the image domain $\Omega$ into regular regions $\Omega_{\text{reg}}$ and occlusion-prone regions $\Omega_{\text{occ}}$, yielding

$$\mathcal{L}(\mathcal{G}) = (1-\kappa)\,\mathcal{L}_{\text{reg}} + \kappa\,\mathcal{L}_{\text{occ}}, \qquad \kappa = \frac{|\Omega_{\text{occ}}|}{|\Omega|}. \tag{3}$$

Since errors are concentrated in $\Omega_{\text{occ}}$, reducing $\mathcal{L}_{\text{occ}}$ delivers the greatest overall improvement, motivating the *refine where it matters* design of GaussUnveil.

## 4 PROPOSED APPROACH

We present GaussUnveil, which generates 3D scenes from surround-view images in a single feed-forward pass. Section 4.1 presents the overall framework of GaussUnveil. Section 4.2 describes the visibility-transition localization. Finally, Section 4.3 details the 3D refinement block architecture. More details of our method are listed in appendix.

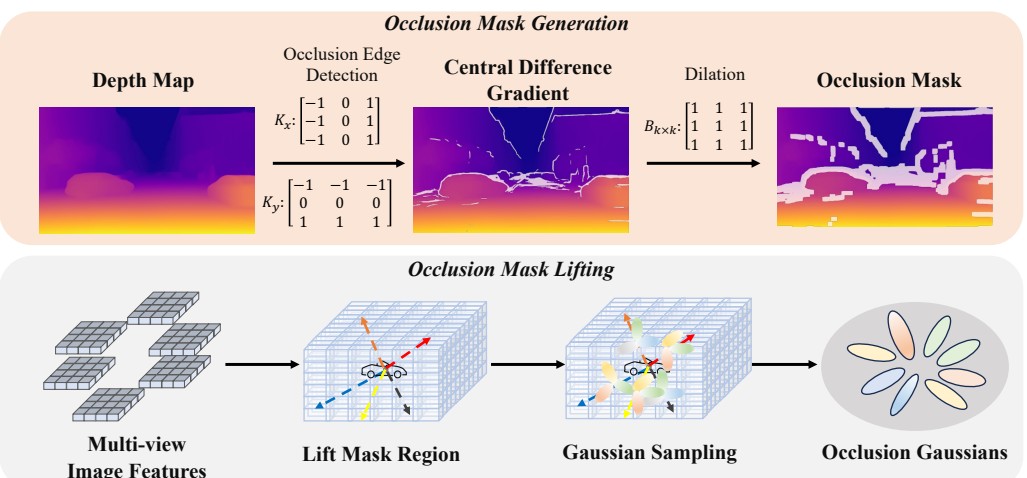

Figure 3: The paradigm of occlusion mask generation and lifting in GaussUnveil. Top: Occlusion mask generation. Depth maps are processed with central-difference gradient filters to detect sharp depth changes, followed by dilation to produce robust occlusion masks. Bottom: Occlusion mask lifting. We lift the mask regions to 3D space with multi-view image features, where Gaussian sampling is performed to instantiate a compact set of occlusion Gaussians that focus refinement on visibility-ambiguous areas.

## 4.1 OVERALL FRAMEWORK

GaussUnveil is a unified reconstruction framework that infers 3D Gaussians from unconstrained viewpoints and performs 3D refinement only to regions likely affected by occlusion. Given multi-view RGB inputs $\{I_i\}_{i=1}^N \in \mathbb{R}^{N \times H \times W \times 3}$, we first extract $4\times$ down-sampled image features $\{F_i\}_{i=1}^N \in \mathbb{R}^{N \times \frac{H}{4} \times \frac{W}{4} \times 3}$ using 2D pretraind image backbones. Then we aim to learn a function $\mathcal{M}$ that maps the down-sampled image features to a set of 3D Gaussians:

$$\mathcal{M} : \{F_i\}_{i=1}^N \rightarrow \left\{(\boldsymbol{\delta}_j, \ \boldsymbol{\alpha}_j, \ \boldsymbol{s}_j, \ \boldsymbol{q}_j, \ \boldsymbol{c}_j)\right\}_{j=1}^K, \tag{4}$$

where $K$ denote the number of 3D Gaussians. $\boldsymbol{\delta}_j, \ \boldsymbol{\alpha}_v, \ \boldsymbol{s}_v, \ \boldsymbol{q}_v,$ and $\boldsymbol{c}_v$ represent the learned offset, opacity, scale, rotation quaternion, and RGB color, respectively.

As shown in Figure 2, we adopt a UNet-style Gaussian encoder–decoder to predict 3D Gaussians from multi-view image features $\{F_i\}_{i=1}^N$, following the design of (Wei et al., 2025). We first upsample the image features and enhance them with Plücker ray encodings and learnable camera embeddings, injecting geometric and view-specific priors. We further concatenate pseudo-depth and its confidence to provide explicit geometry. The resulting features are processed with a stack of downsampling blocks, a bottleneck block, and symmetric upsampling blocks with skip connections, enabling hierarchical context aggregation. These blocks utilized patchified cross attentions for efficient cross-view correlation. These aggregation features are fed into several convolution layers to obtain the per-pixel depth and 3D Gaussians. To compute the center $\boldsymbol{\mu}_p$, we first unproject the pixel from the ray origin $\boldsymbol{o}_p$ along the ray direction $\boldsymbol{r}_p$ using the depth $\boldsymbol{d}_p$, then refine this coarse position with the learned offset $\boldsymbol{\delta}_p \in \mathbb{R}^3$, represented as

$$\boldsymbol{\mu}_p \ = \ \boldsymbol{o}_p \ + \ \boldsymbol{d}_p \, \boldsymbol{r}_p \ + \ \boldsymbol{\delta}_p. \tag{5}$$

Throughout the above steps, we obtain the pixel-based Gaussians $\left\{(\boldsymbol{\delta}_j, \ \boldsymbol{\alpha}_j, \ \boldsymbol{s}_j, \ \boldsymbol{q}_j, \ \boldsymbol{c}_j))\right\}_{j=1}^{K_p}$. Although pixel-based Gaussians can reconstruct most regions effectively, our experiments and theoretical analysis reveal that potentially occluded areas in multi-view images introduce ambiguities during rendering and lead to performance degradation. Unlike the previous dual-branch OmniScene (Wei et al., 2025), we explicitly localize these potentially occluded regions by a visibility transition localization module (Section 4.2) and further refine the Gaussians from these occlusion regions with a 3D Gaussian refinement module (Section 4.3). This strategy not only preserves reconstruction performance but also significantly reduces the number of Gaussians, thereby improving inference efficiency.

## 4.2 VISIBILITY TRANSITION LOCALIZATION

Pixel unprojection accounts for most rays in sparse-view reconstruction, while large errors arise at visibility transitions where sightlines shift between foreground and background. To address this, we detect such transitions, expand them into an occlusion mask, lift the mask into 3D, and refine only within the masked regions shown in Figure 3.

**Occlusion Mask Generation.** In surround-view images, the depth at each pixel is the distance to the first surface along its camera ray. When the same surface stays visible as the pixel moves a little, depth changes smoothly. At an occlusion boundary, the frontmost surface switches from the foreground to the background, leading to an abrupt change in depth. We then utilize finite differences at this switch yields large depth gradients, which can detect occlusion boundaries effectively. Specifically, for the multi-view images with the corresponding predicted depth $Z_i$, we compute central-difference gradients as

$$D_x = Z_i \times K_x, \ D_y = Z_i \times K_y, \ E_i = \sqrt{D_x^2 + D_y^2}, \tag{6}$$

with $K_x = [-1, 0, 1]$, $K_y = [-1, 0, 1]^\top$ and the operator $\times$ denote the 2D convolution. We further introduce a hyperparameter $\tau_g$ to threshold the computed difference gradients to obtain the boundary between foreground and background,

$$\mathcal{O}_b(x,y) = \begin{cases} 1, & \text{if } E_i(x,y) \geq \tau_g \\ 0, & \text{otherwise} \end{cases} \tag{7}$$

where $\{x, y\}$ denote the corresponding coordinates of the image plane. Subsequently, we utilize morphological dilation with a square structuring element to expand these boundaries into uncertainty bands, yielding an occlusion mask that localizes likely occluded or geometry-discontinuous regions. In generally, a square structuring window of size $k$ (radius $r = \frac{k-1}{2}$), we obtain the dilated mask $\mathcal{O}_d$ as

$$\mathcal{O}_d(x,y) \ = \ \mathbf{1}\left(\sum_{i=-r}^{r} \sum_{j=-r}^{r} E(x+i, \, y+j) \ > \ 0\right), \tag{8}$$

where $\mathbf{1}(\cdot)$ is the indicator function. This means that if there is at least one pixel within the $k \times k$ neighborhood centered at $(x, y)$, the dilation result at that position is set to 1. In this way, the original boundaries are expanded into a wider occlusion mask region.

**Occlusion Mask Lifting.** Given per-view occlusion masks $\mathcal{O}_d \in \{0, 1\}^{H \times W}$, we lift them to a thin 3D neighborhood around depth discontinuities, which we call the *visibility transition tube* $\mathcal{O}_{3d} \subset \mathbb{R}^3$. We project the occluded pixels into rays in the camera coordinate system and further transform them into rays in the world coordinate system. For each potentially occluded pixel, we take its predicted depth $\mathbf{d_{center}}$ as the center of a line segment and then compute the near and far endpoints $[\mathbf{p_0}, \mathbf{p_1}]$ as

$$\begin{aligned} \mathbf{p_0} &= \mathbf{o}_p \ + \ (\mathbf{d_{center}} \ - \ \boldsymbol{\delta}_p) \, \mathbf{d}_p, \\ \mathbf{p_1} &= \mathbf{o}_p \ + \ (\mathbf{d_{center}} \ + \ \boldsymbol{\delta}_p) \, \mathbf{d}_p. \end{aligned} \tag{9}$$

Here, $\boldsymbol{\delta}_p$ denotes the longitudinal thickness along the ray, which can be adjusted according to scale or uncertainty, and can be expressed as

$$\boldsymbol{\delta}_p(\boldsymbol{u}) = \kappa_{\text{rel}} \cdot \mathbf{d_{center}} \ + \ \kappa_{\text{abs}}, \tag{10}$$

where $\kappa_{\text{rel}}$ and $\kappa_{\text{abs}}$ are two hyperparameters used to control the longitudinal length along the camera ray, allowing the thickness to increase linearly with depth: the farther the point, the greater the permitted longitudinal uncertainty. We assume that sampling along the line segment $[\mathbf{p_0}, \mathbf{p_1}]$ can effectively cover the 3D spatial position of the occluded point. The 3D tube $\mathcal{O}_{3d}$ is the union of all lifted segments across views,

$$\mathcal{O}_{3d} = \bigcup_{v \in \mathcal{V}} \ \bigcup_{\mathcal{O}_v(x,y)=1} \mathcal{S}_{x,y}^v, \tag{11}$$

where $\mathcal{S}_{x,y}^v$ denote the 3D segment at position $\{x, y\}$ in different views $v$.

### 4.3 3D GAUSSIAN REFINEMENT

Pixel-based Gaussians already reconstruct most of the scene under sparse views, but they break down near visibility changes where geometry is hidden or truncated. Motivated by (Huang et al., 2024), we propose a lightweight 3D Gaussian refinement block to refine these Gaussians using the visibility transition tube from Section 4.2. We first initialize the Gaussian queries as learnable vectors. Then, we iteratively refine the Gaussians within $N$ 3D Gaussian Refinement blocks. Each block consists of a self-context aggregation layer to aggregate the context information of Gaussian queries, a cross-view aggregation layer to aggregate visual cues from different views, and a corrective refinement layer to rectify the properties of 3D Gaussians.

**Self-context Aggregation.** We utilize 3D sparse convolution (Contributors, 2022) to build our self-context aggregation layer. We voxelize the point represented by the center of each Gaussian and then perform 3D sparse convolution on the occupied voxels only. Since the number of Gaussians is far smaller than the dense grid size $X \times Y \times Z$, this operation avoids the cubic cost of dense 3D processing. The range of receptive can be expanded by stacking multiple layers of sparse convolution. To maintain the spatial sparsity, we use one 3D convolution in a self-context aggregation layer.

**Cross-view Aggregation.** We introduce the cross-view aggregation layer to enrich these Gaussian queries with cross-view context. Specifically, for a 3D Gaussian query $Q_{3d}$, we perform deformable attention (DA) (Zhu et al., 2020) onto multi-view image feature maps to aggregate visual cues from different views. Cross-view aggregation effectively addresses occlusions that occur in single views, as it allows each Gaussian to acquire complementary features from multiple viewpoints.

**Corrective Refinement.** The goal of the corrective refinement layer is to rectify the Gaussian properties with the corresponding Gaussian queries updated from self-context aggregation and cross-view aggregation layers. Specifically, we utilize a multi-layer perceptron (MLP) to decode the updated Gaussian properties from the Gaussian queries. Notably, we refine the mean of each Gaussian through a residual structure, while the other properties are directly replaced by their updated values.

We can obtain a compact, occlusion-aware Gaussian representation in 3D space by stacking several 3D Gaussian refinement blocks. Compared with pixel-based Gaussians, this representation mitigates boundary ambiguities arising from occlusions and improves multi-view consistency, leading to more robust reconstructions.

## 5 EXPERIMENTS

### 5.1 EXPERIMENTAL SETTINGS

**Evaluation Tasks.** We follow the experimental protocol of OmniScene (Wei et al., 2025) and evaluate GaussUnveil in two settings: the ego-centric setting on nuScenes (Caesar et al., 2020) and the scene-centric setting on RealEstate10K (Zhou et al., 2018). For both datasets, we compare against the 3DGS-based methods OmniScene (Wei et al., 2025), pixelSplat (Charatan et al., 2024), and MVSplat (Chen et al., 2024), as well as the light-field approach AttnRend (Du et al., 2023) and the NeRF-based method MuRF (Xu et al., 2024). Additional details are provided in the appendix.

**Metrics.** We adopt three widely used metrics from prior reconstruction studies (Wei et al., 2025; Chen et al., 2024; Charatan et al., 2024) to evaluate visual quality: peak signal-to-noise ratio (PSNR), structural similarity index (SSIM) (Wang et al., 2004), and learned perceptual image patch similarity (LPIPS) (Zhang et al., 2018). Higher values indicate better performance for PSNR and SSIM, whereas lower values are preferred for LPIPS. In addition, we report the Pearson correlation coefficient (PCC) (Sedgwick, 2012) to assess the geometric fidelity of reconstructed 3D scenes.

**Implementation Details.** We implement GaussUnveil in PyTorch using the open-source Gaussian renderer (Kerbl et al., 2023). Multi-view image features are extracted with a ResNet-50 backbone pre-trained using DINO (Caron et al., 2021). For occlusion mask generation, depth values are clipped to the range $[0, 100]$, with the dilation kernel size as 7 and threshold as 3. We employ four Gaussian refinement blocks to update Gaussians within the visibility transition tube. Training is performed on two NVIDIA A800 GPUs for 100k iterations with a batch size of 4 on nuScenes (Caesar et al., 2020), and on a single A800 GPU for 300k iterations with a batch size of

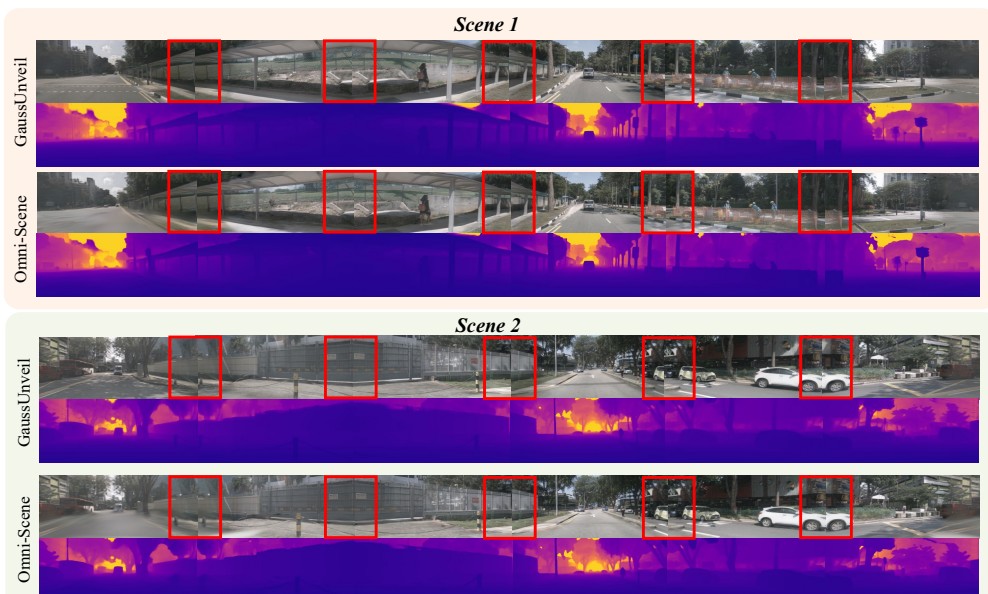

Figure 4: The qualitative comparison of reconstruction performance between Omni-Scene (Wei et al., 2025) and our GaussUnveil (better viewed when zoomed in). We render six views to cover the full 360° panorama, ensuring approximately 15% overlap between adjacent viewpoints. The red boxes indicate the overlapping regions across different views.

Table 1: Quantitative results of the ego-centric reconstruction task on nuScenes (Caesar et al., 2020). PCC is reported as N/A for AttnRend (Du et al., 2023), since it does not produce an interpretable 3D structure for depth rendering.

| Method | Time(s) | Param(M) | PSNR↑ | SSIM↑ | LPIPS↓ | PCC↑ |
|---|---|---|---|---|---|---|
| AttnRend (Du et al., 2023) | 9.98 | 125.1 | 20.96 | 0.533 | 0.467 | N/A |
| MuRF (Xu et al., 2024) | 0.672 | **5.3** | 20.34 | 0.504 | 0.433 | -0.332 |
| pixelSplat (Charatan et al., 2024) | 0.508 | 125.4 | 21.51 | 0.616 | 0.372 | 0.001 |
| MVSplat (Chen et al., 2024) | 0.174 | 12.0 | 21.61 | 0.658 | 0.295 | 0.181 |
| OmniScene (Wei et al., 2025) | 0.088 | 81.7 | 24.27 | 0.736 | 0.237 | 0.800 |
| GaussUnveil (ours) | **0.058** | 80.3 | **24.65** | **0.754** | **0.220** | **0.837** |

8 on RealEstate10K (Zhou et al., 2018). Optimization uses AdamW (Kingma & Ba, 2014) with an initial learning rate of $1 \times 10^{-4}$ and cosine decay. More details are provided in the appendix.

## 5.2 MAIN RESULTS

**Results on nuScenes.** Table 1 presents a comparison between GaussUnveil and existing baselines on the nuScenes dataset. Compared to the state-of-the-art Omni-Scene (Wei et al., 2025), specifically designed for the ego-centric setting, our approach is ≈34% faster while also achieving higher accuracy. Feed-forward sparse-view methods (Chen et al., 2024; Charatan et al., 2024; Xu et al., 2024; Du et al., 2023) perform worst, particularly on the PCC metric, as limited view overlap in ego-centric settings makes depth estimation unreliable. While Omni-Scene improves over MVSplat and PixelSplat, its voxel-based Gaussian branch has millions of primitives, even in well-observed regions where pixel-based Gaussians suffice. In contrast, our method targets refinement only to occluded regions, substantially reducing Gaussian count while preserving performance. Qualitative results on nuScenes (Figure 4) further show that GaussUnveil achieves reconstructions on par with OmniScene while operating more efficiently.

**Results on RealEstate10K.** To further demonstrate the effectiveness and generalization of our proposed method, we also conduct evaluations on the RealEstate10K (Zhou et al., 2018) dataset, a scene-centric benchmark widely used for sparse-view reconstruction tasks.

As shown in Figure 2, GaussUnveil achieves the best performance on SSIM and LPIPS metrics. We also note that feed-forward baselines, such as pixelSplat (Charatan et al., 2024) and MuRF (Xu et al., 2024), although efficient, suffer from limited geometric fidelity, particularly in terms of PCC. The comparison between ego-centric methods, such as OmniScene, and GaussUnveil underscores the effectiveness of the proposed *refine where it matters* strategy.

Table 2: Quantitative results of RealEstate10K (Zhou et al., 2018) under scene-centric reconstruction setting.

| Method | PSNR↑ | SSIM↑ | LPIPS↓ | PCC↑ |
|---|---|---|---|---|
| AttnRend (Du et al., 2023) | 24.78 | 0.820 | 0.213 | N/A |
| MuRF (Xu et al., 2024) | 26.10 | 0.858 | 0.143 | 0.344 |
| pixelSplat (Charatan et al., 2024) | 25.89 | 0.858 | 0.142 | 0.285 |
| MVSplat (Chen et al., 2024) | **26.39** | 0.869 | 0.128 | 0.363 |
| OmniScene (Wei et al., 2025) | 26.19 | 0.865 | 0.131 | **0.368** |
| **GaussUnveil (ours)** | 26.32 | **0.872** | **0.123** | 0.365 |

### 5.3 ABLATION STUDY

**Effectiveness of Occlusion-aware Refinement.** We conduct ablations to evaluate the effectiveness of our core contribution, the Occlusion-aware Refinement. 'w/o Refinement' denotes retaining only the backbone network without refining potentially occluded regions, making the structure similar to the pixel-based Gaussian branch in Omni-Scene. 'w/o. Depth Init' indicates that our method does not use depth from Metric3D (Yin et al., 2023) for initialization. 'w/o. Mask' means we still perform Gaussian refinement, but instead of initializing Gaussians in potentially occluded regions, we randomly select their positions. As shown in Table 3, we find that removing any of these components leads to performance degradation. We observe that eliminating the occlusion-aware refinement significantly degrades performance, with PSNR dropping to 22.89 and SSIM to 0.698. This highlights the importance of selectively refining occluded regions, as the backbone alone struggles to handle visibility ambiguities. We also note that removing depth initialization leads to a notable decline in PCC, indicating that depth maps are crucial for geometric structure. Finally, performing refinement without occlusion masks, i.e., randomly seeding Gaussians, yields the worst overall results, demonstrating that targeted seeding in ambiguous regions is key to both reconstruction accuracy and perceptual quality.

Table 3: Ablation study on ego-centric reconstruction on nuScenes (Caesar et al., 2020).

| Method | PSNR↑ | SSIM↑ | LPIPS↓ | PCC↑ |
|---|---|---|---|---|
| w/o. Refinement | 22.89 | 0.698 | 0.290 | 0.780 |
| w/o Depth Init | 24.41 | 0.743 | 0.226 | 0.654 |
| w/o. Mask | 21.40 | 0.654 | 0.306 | 0.720 |
| w/o. SA | 24.04 | 0.738 | 0.234 | 0.827 |
| w/o. CA | 23.30 | 0.723 | 0.265 | 0.802 |
| **Ours** | **24.65** | **0.754** | **0.220** | **0.837** |

**Effectiveness of 3D Gaussian Refinement.** We further conduct ablation studies on the 3D Gaussian Refinement Block. We remove the self-context aggregation and cross-view aggregation layers, denoted as 'w/o SA' and 'w/o CA', respectively. The corrective refinement layer, which is responsible for decoding the updated features of Gaussians, cannot be ablated. As shown in Table 3, removing either aggregation module results in a noticeable performance drop. Moreover, our analysis reveals that the cross-view aggregation layer has a stronger impact on reconstruction quality compared to the self-context aggregation layer. This is because refined Gaussians iteratively obtain features both from neighboring Gaussians and from multi-view image features, and the information carried by multi-view features is substantially richer.

## 6 CONCLUSION

In this work, we introduced GaussUnveil, an occlusion-aware selective-refinement framework for sparse-view ego-centric 3D reconstruction. By unveiling regions of uncertainty through depth-gradient masks and restricting refinement to occlusion-prone areas, GaussUnveil shifts the paradigm from *refining everywhere* to *refining where it matters*. Our lightweight refinement block effectively updates Gaussians with self-context and multi-view features, while a mask-aware objective stabilizes training around visibility boundaries. Experiments on both ego-centric and scene-centric benchmarks confirm that GaussUnveil achieves superior reconstruction quality with significantly fewer Gaussians compared to Omni-Scene. These results highlight that targeted refinement, rather than uniform processing, provides a more efficient pipeline for 3D scene reconstruction.

## 7 ETHICS STATEMENT

This research adheres to the ethical guidelines of the ICLR community. Our work focuses on developing machine learning methods for 3D scene reconstruction and does not involve collection of sensitive personal information or data that may compromise individual privacy. All datasets used in this study are publicly available benchmark datasets, such as nuScenes and RealEstate10K, that have been released under appropriate licenses for research purposes. We carefully ensured compliance with dataset usage policies and did not perform any data manipulation that would raise ethical concerns. Potential societal impacts of our work include both positive and negative aspects. On the positive side, our method may advance the state-of-the-art in autonomous driving, potentially improving safety and efficiency. On the negative side, there exists the possibility of misuse in surveillance or military applications. We acknowledge these risks and emphasize that our work is intended solely for academic research and beneficial applications. No human subjects, personally identifiable information, or harmful synthetic content were involved in this study. We believe the ethical risks of this work are minimal and manageable.

## 8 REPRODUCIBILITY STATEMENT

We are committed to ensuring the reproducibility of our results, in accordance with the ICLR reproducibility guidelines. We will release the core code upon publication. All datasets used in our experiments are publicly available, including nuScenes and RealEstate10K. We provide complete details of training hyperparameters (learning rate, batch size, optimizer, weight decay, training epochs, etc.) in our draft. Detailed descriptions of our architecture, including layer configurations and parameter counts, are reported in the Method section and appendix. Experiments were conducted on NVIDIA A800 GPUs, and we report speed and model size. We believe these measures are sufficient for independent researchers to fully reproduce our results.

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

# A APPENDIX

## A.1 USE OF LARGE LANGUAGE MODELS (LLMS)

In preparing this manuscript, we made limited use of large language models (LLMs), specifically OpenAI ChatGPT, to assist with improving the clarity and style of the writing. The scientific content, experimental design, theoretical derivations, and results were conceived, implemented, and validated entirely by the authors. LLMs were not used for generating novel scientific ideas, experiments, or analyses. All outputs from LLMs were carefully reviewed, verified, and edited by the authors to ensure correctness and originality. No proprietary or unpublished data were provided to LLMs during manuscript preparation. All datasets, code, and results reported in this paper are entirely the work of the authors.

## A.2 METHOD ANYLASIS

### A.2.1 WHY *Refine Where It Matters* PARADIGM WORKS?

In sparse-view settings, most pixels are well-explained by pixel-based models, while uncertainty concentrates near occlusion boundaries. Updating all regions wastes gradient budget on *easy* pixels with weak or noisy signals, often irrelevant to true errors. Restricting updates to uncertainty regions ensures that (i) samples target where improvement is needed, (ii) reliable regions do not bias optimization, and (iii) gradient signal-to-noise is maximized for the same compute.

Let $\mathcal{D} = \mathcal{U} \cup \mathcal{G}$ denote the set of uncertain and good pixels with proportions $\pi_U$ and $\pi_G = 1 - \pi_U$. For each pixel $p$, the gradient is $g(p) = \nabla_\Theta \ell_p$, with group means $\mu_U = \mathbb{E}[g \mid \mathcal{U}]$, $\mu_G = \mathbb{E}[g \mid \mathcal{G}]$ and covariances $\Sigma_U, \Sigma_G$. We then optimize the masked objective as

$$\mathcal{L}_U(\Theta) = \mathbb{E}[\ell_p(\Theta) \mid p \in \mathcal{U}]. \tag{12}$$

With minibatch size $B$, the estimators of *refine everywhere* (RE) and *refine where it matters* (RWM) satisfy $\mathbb{E}[\widehat{g}_{\mathrm{RWM}}] = \mu_U$ and $\mathbb{E}[\widehat{g}_{\mathrm{RE}}] = \pi_U \mu_U + \pi_G \mu_G$. Thus RWM is unbiased for the *desired* descent direction of $\mathcal{L}_U$, while RE estimates a mixture mean. For the same pixel budget, we define the variance as

$$\mathrm{Var}(\widehat{g}_{\mathrm{RWM}}) = \tfrac{1}{B}\Sigma_U, \qquad \mathrm{Var}(\widehat{g}_{\mathrm{RE}}) = \tfrac{1}{B}(\pi_U \Sigma_U + \pi_G \Sigma_G) + \pi_U \pi_G (\mu_U - \mu_G)(\mu_U - \mu_G)^\top. \tag{13}$$

RE not only allocates effectively $B\pi_U$ samples to $\mathcal{U}$ (worse SNR by $1/\pi_U$), but also incurs an irreducible *mixture-bias* term. We assume that $\mathcal{L}_U$ is $L-$smooth and tak one step $\Theta^+ = \Theta - \eta\,\widehat{g}$ with $\eta \le 1/L$,

$$\mathbb{E}[\mathcal{L}_U(\Theta^+)] \le \mathcal{L}_U(\Theta) - \eta \underbrace{\langle \nabla \mathcal{L}_U(\Theta), \mathbb{E}[\widehat{g}] \rangle}_{\text{alignment \& signal}} + \tfrac{L\eta^2}{2}\underbrace{\mathbb{E}\|\widehat{g}\|^2}_{\text{noise}}. \tag{14}$$

Under the assumption $\langle \mu_G, \mu_U \rangle \le 0$ and $\|\mu_G\| \ll \|\mu_U\|$ (good pixels need little refinement), we obtain

$$\begin{aligned} \Delta_{\mathrm{RWM}} &\le -\eta\|\mu_U\|^2 + \tfrac{L\eta^2}{2}\Big(\|\mu_U\|^2 + \tfrac{\mathrm{tr}\,\Sigma_U}{B}\Big) < \\ &-\eta\,\pi_U\|\mu_U\|^2 + \tfrac{L\eta^2}{2}\Big(\|\pi_U\mu_U + \pi_G\mu_G\|^2 + \tfrac{\pi_U\,\mathrm{tr}\,\Sigma_U + \pi_G\,\mathrm{tr}\,\Sigma_G}{B}\Big) \le \Delta_{\mathrm{RE}}, \end{aligned} \tag{15}$$

for sufficiently large $B$ (or equivalently small $\eta$). The left inequality reflects perfect alignment and higher useful-sample allocation of RWM; the right inequality follows from the extra mixture magnitude/variance in RE (cf. equation 13).

The above analysis shows that for the same compute, masking to uncertainty regions produces a gradient that is (i) unbiased for the target descent direction, (ii) higher SNR by $\approx 1/\pi_U$, and (iii) yields a strictly larger expected one-step loss decrease. Hence *refine where it matters* is easier to optimize and more compute-efficient than *refine everywhere*.

### A.2.2 WHY *Visibility Transition Localization* MODULE WORKS?

In this part, we provide the theoretical analysis of our proposed *Visibility Transition Localization* module. Let a calibrated pinhole camera with intrinsics $(f_x, f_y)$ and projection $\pi : \mathbb{R}^3 \to \mathbb{R}^2$. For

a visible surface $S$ with depth function $z : \Omega \subset \mathbb{R}^2 \to \mathbb{R}_+$, define disparity $d(u) = fB/z(u)$ (for stereo baseline $B$ or any inverse-depth proxy; analysis is identical with $1/z$). In piecewise-smooth regions with a single visible surface, $d$ is $\mathcal{C}^1$ and satisfies (e.g., standard shape-from-shading geometry) $\|\nabla d(u)\| \leq K_{\text{surf}}$ (bounded by surface curvature and foreshortening). At occluding contours (depth/visibility transitions), the frontmost surface changes discontinuously along the ray: the visibility indicator $\chi(z)$ (front-most along the ray) is a step function in $z$, and $d(u)$ is piecewise-smooth with jump discontinuities. Formally, along any scanline $\gamma(t)$ crossing an occlusion at $t_0$,

$$d(\gamma(t)) = \begin{cases} d_1(t) & t < t_0, \\ d_2(t) & t > t_0, \end{cases} \qquad d_1, d_2 \in \mathcal{C}^1, \quad \lim_{t\uparrow t_0} d_1(t) \neq \lim_{t\downarrow t_0} d_2(t). \tag{16}$$

Hence $d$ is a bounded variation (BV) function with distributional derivative

$$\nabla d = \nabla d_{\text{ac}} + \nu\, \delta_\Gamma, \tag{17}$$

where $\nabla d_{\text{ac}}$ is absolutely continuous (bounded in smooth regions), $\Gamma$ is the occluding contour, $\delta_\Gamma$ is a 1-D Dirac measure supported on $\Gamma$, and $\nu$ is the jump magnitude.

Let $G_h$ be a finite-difference gradient operator at pixel spacing $h$. Then for pixels $u$ not on $\Gamma$,

$$\|G_h d(u)\| \leq K_{\text{surf}} + \mathcal{O}(h), \tag{18}$$

while for pixels whose stencil intersects $\Gamma$,

$$\|G_h d(u)\| \geq \frac{|\nu|}{h} - \mathcal{O}(1), \tag{19}$$

i.e., the discrete gradient blows up as $h \to 0$ at visibility transitions.

Edges localize the 2D projection of visibility transitions while true uncertainty extends slightly around them because of calibration noise, quantization, and coarse 3D initialization. We therefore dilate edges by a radius that upper-bounds pixel-space uncertainty. Let $u = \pi(X)$ and let the total 2D localization error be

$$r \geq \underbrace{\left\|\frac{\partial \pi}{\partial X}\right\| \sigma_X}_{\text{3D init error}} + \underbrace{\sigma_{\text{cal}}}_{\text{calibration}} + \underbrace{\sigma_{\text{disc}}}_{\text{discretization}} + \underbrace{\sigma_{\text{noise}}}_{\text{sensor}}. \tag{20}$$

We define the uncertainty region as the morphological dilation

$$\mathcal{B} = \{u : \text{dist}(u, \Gamma) \leq r\} = E \oplus \mathbb{B}_r. \tag{21}$$

A first-order perturbation moves the projected edge by at most $r$ in pixels. Dilation by $r$ covers all projections under the bounded perturbation model; hence any 3D point whose visibility is ambiguous projects inside $\mathcal{B}$. Thresholding removes smooth regions; dilation can only admit pixels within distance $r$ of detected edges. If $r$ is smaller than the separation to other (non-occluding) edges, they remain excluded.

## A.3 ADDITIONAL EXPERIMENTS

### A.3.1 EXPERIMENTAL SETTINGS

**Dataset.** For the ego-centric setting, we evaluate GaussUnveil on nuScenes (Caesar et al., 2020) following OmniScene (Wei et al., 2025). There are 700 training scenes and 150 validation scenes in nuScenes are divided into uniformly spaced bins along the vehicle trajectory. In each bin, the first and last frames are 3.2 m apart. The center frame provides six surround-view images as input views, and the first and last frames provide twelve images as target novel views. We use 135,941 bins for training and 30,080 bins for validation, with an image resolution of $224 \times 400$. To compare with prior feed-forward reconstruction methods, we also conduct evaluations on RealEstate10K (Zhou et al., 2018), a large scene-centric dataset with indoor and outdoor scenes under the scene-centric setting. RealEstate10K (Zhou et al., 2018)is collected from in-the-wild YouTube videos of real estate tours. It contains approximately 10,000 videos, from which multi-view image sequences with associated camera poses are extracted. The dataset covers a wide variety of indoor scenes with diverse layouts and lighting conditions, making it a standard benchmark for novel view synthesis

and scene-centric 3D reconstruction. Following the protocol in prior work (Wei et al., 2025; Chen et al., 2024; Charatan et al., 2024), we use 67,477 scenes for training and 7,289 scenes for testing.

**Metrics.** We use PSNR, SSIM, LPIPS and PCC metrics to evaluate the performance of our method. PSNR measures pixel-level fidelity based on mean squared error (MSE) as

$$\text{PSNR} = 10 \cdot \log_{10}\left(\frac{\text{MAX}_I^2}{\text{MSE}}\right), \quad \text{MSE} = \frac{1}{HW}\sum_{i=1}^{H}\sum_{j=1}^{W}(I_{ij} - \hat{I}_{ij})^2, \quad (22)$$

where $\text{MAX}_I$ is the maximum pixel value, and $I, \hat{I}$ denote the ground-truth and reconstructed images. Higher PSNR indicates better low-level fidelity. SSIM evaluates perceptual quality by comparing luminance, contrast, and structure as

$$\text{SSIM}(x,y) = \frac{(2\mu_x\mu_y + C_1)(2\sigma_{xy} + C_2)}{(\mu_x^2 + \mu_y^2 + C_1)(\sigma_x^2 + \sigma_y^2 + C_2)}, \quad (23)$$

where $\mu_x, \mu_y$ are means, $\sigma_x^2, \sigma_y^2$ are variances, and $\sigma_{xy}$ is covariance. $C_1, C_2$ are constants to stabilize the division. LPIPS measures perceptual distance using deep features $\phi_l(\cdot)$ from pretrained networks as

$$\text{LPIPS}(x,y) = \sum_l \frac{1}{H_l W_l} \sum_{h,w} \left\| w_l \odot \left( \phi_l(x)_{h,w} - \phi_l(y)_{h,w} \right) \right\|_2^2, \quad (24)$$

where $w_l$ are learned weights for each feature channel. Lower LPIPS values correspond to reconstructions that are perceptually closer to human judgments. PCC measures linear correlation between predicted geometry $X$ and ground truth $Y$ as

$$\text{PCC}(X,Y) = \frac{\sum_{i=1}^{N}(X_i - \bar{X})(Y_i - \bar{Y})}{\sqrt{\sum_{i=1}^{N}(X_i - \bar{X})^2}\sqrt{\sum_{i=1}^{N}(Y_i - \bar{Y})^2}}, \quad (25)$$

where $\bar{X}, \bar{Y}$ are the means. Values closer to 1 indicate stronger geometric consistency.

**Implementation Details.** For the 2D image encoder, we adopt a ResNet-50 backbone pre-trained with DINO, and employ a feature pyramid network (FPN) with the P2 level for feature extraction. The extracted multi-view features are subsequently fed into the reconstruction pipeline. The pixel-based Gaussian predictor is configured with four downsampling and four upsampling stages. The channel dimensions for the downsampling path are set to $\{128, 256, 512, 512\}$, while the upsampling path mirrors this structure with $\{512, 512, 256, 128\}$. Correspondingly, the number of patches per stage is $\{8, 8, 4, 2\}$ for downsampling and $\{2, 4, 8, 8\}$ for upsampling. This design allows for multiscale feature aggregation across views. Then, we utilize three convolutional layers, which decode the fused features into pixel-aligned Gaussians. For training, we adopt the Adam optimizer with $\beta_1 = 0.9$ and $\beta_2 = 0.999$, a weight decay of 0.01, and a cosine learning rate scheduler. The model is trained for 100k iterations with an initial learning rate of $1 \times 10^{-4}$. A warm-up phase of 1000 iterations is used, and gradient clipping is applied with a maximum norm of 1.0 to stabilize optimization.

**Rendering Settings for Visualization.** We generate a 360° sweep of six in-place yaw views by rotating a base pose $\mathbf{T}_{\text{base}} \in SE(3)$ (with $\mathbf{R}_{\text{base}} \in SO(3)$, $\mathbf{t}_{\text{base}} \in \mathbb{R}^3$) about its local $y$-axis while fixing the optical center. Let the horizontal FoV be $\phi_x$ (radians) and define the uniform step

$$\Delta\theta = \frac{2\pi}{n}, \qquad n = 6. \quad (26)$$

To obtain an adjacent horizontal overlap $\kappa \in (0,1)$, we match 1D angular coverage,

$$\kappa \approx 1 - \frac{\Delta\theta}{\phi_x} \quad \Rightarrow \quad \phi_x \approx \frac{2\pi}{n(1-\kappa)}. \quad (27)$$

With $\kappa = 0.15$ and $n = 6$,

$$\phi_x^\star = \frac{2\pi}{6(1 - 0.15)} = \frac{2\pi}{5.1}. \quad (28)$$

Given a base FoV $\phi_x^{(0)}$, we use $\phi_x = \max\left(\phi_x^{(0)}, \phi_x^\star\right)$ (vertical FoV $\phi_y$ remains $\phi_y^{(0)}$). Yaw angles are

$$\theta_i = i\,\Delta\theta, \qquad i = 0, \ldots, n-1, \quad (29)$$

with local rotation

$$\mathbf{R}_y(\theta) = \begin{bmatrix} \cos\theta & 0 & \sin\theta \\ 0 & 1 & 0 \\ -\sin\theta & 0 & \cos\theta \end{bmatrix}, \qquad \mathbf{T}_y(\theta) = \begin{bmatrix} \mathbf{R}_y(\theta) & \mathbf{0} \\ \mathbf{0}^\top & 1 \end{bmatrix}, \tag{30}$$

and synthesized poses

$$\mathbf{T}_i = \mathbf{T}_{\text{base}}\mathbf{T}_y(\theta_i), \tag{31}$$

which rotate the camera in place. Using $\phi_x^{(i)} \equiv \phi_x$ and $\phi_y^{(i)} \equiv \phi_y^{(0)}$, we render Gaussians $\mathcal{G}$ as

$$(\mathbf{I}_i, \mathbf{D}_i) = \text{Render}(\mathcal{G}, \mathbf{T}_i, \phi_x^{(i)}, \phi_y^{(i)}), \qquad i = 0, \ldots, 5, \tag{32}$$

yielding six evenly spaced views around the circle with $\approx 15\%$ horizontal overlap.

### A.3.2 MORE EXPERIMENTS

Table 4: The different settings of GaussUnveil on nuScenes. We report PSNR, SSIM, LPIPS and PCC metrics.

(a) **Gaussian numbers**. The best Gaussian Nums is 10000.

| Nums | PSNR | SSIM | LPIPS | PCC |
|---|---|---|---|---|
| 5000 | 23.79 | 0.747 | 0.231 | 0.830 |
| 10000 | 24.65 | 0.753 | 0.220 | 0.837 |
| 20000 | **24.67** | **0.756** | **0.218** | **0.842** |

(b) **Refinement blocks number.** Four works best.

| Nums | PSNR | SSIM | LPIPS | PCC |
|---|---|---|---|---|
| 1 | 23.76 | 0.747 | 0.224 | 0.828 |
| 4 | **24.65** | **0.753** | **0.220** | **0.837** |
| 6 | 24.56 | 0.744 | 0.226 | 0.841 |

(c) **Dilation kernel size.** Seven works best.

| Size | PSNR | SSIM | LPIPS | PCC |
|---|---|---|---|---|
| 1 | 23.78 | 0.736 | 0.232 | 0.835 |
| 3 | 24.13 | 0.750 | 0.226 | 0.832 |
| 7 | **24.65** | **0.753** | **0.220** | **0.837** |

**Additional ablations.** We conduct several ablations on different settings of GaussUnveil on nuScenes. Table 4 presents ablations on the number of Gaussians, the number of refinement blocks, and the dilation kernel size for occlusion mask generation, evaluated on nuScenes. We vary the number of initial Gaussians from 5k to 20k. Performance steadily improves with more Gaussians, peaking at 20k (PSNR 24.67, SSIM 0.756, LPIPS 0.218, PCC 0.842). However, the gap between 10k and 20k is marginal, while 10k maintains a lower memory footprint. Thus, we adopt 10k as the default. We test between 1 and 7 refinement blocks. Using only one block underfits (PSNR 23.76, LPIPS 0.224), while stacking four blocks achieves the best trade-off (PSNR 24.65, SSIM 0.753, LPIPS 0.220, PCC 0.837). Increasing to seven blocks brings no further benefit, suggesting diminishing returns with deeper refinement. For occlusion mask generation, we vary the dilation kernel size from 1 to 7. A kernel size of 1 yields poor SSIM and LPIPS due to under-coverage of uncertainty regions. A kernel size of 7 achieves the best overall performance (PSNR 24.65, SSIM 0.753, LPIPS 0.220, PCC 0.837), while excessively large kernels risk including irrelevant pixels.

**More Visualizations.** We also provide more qualitative comparisons of reconstruction performance with other methods. We can observe that our GaussUnveil also achieves promising reconstruction quality using fewer additional Gaussians.

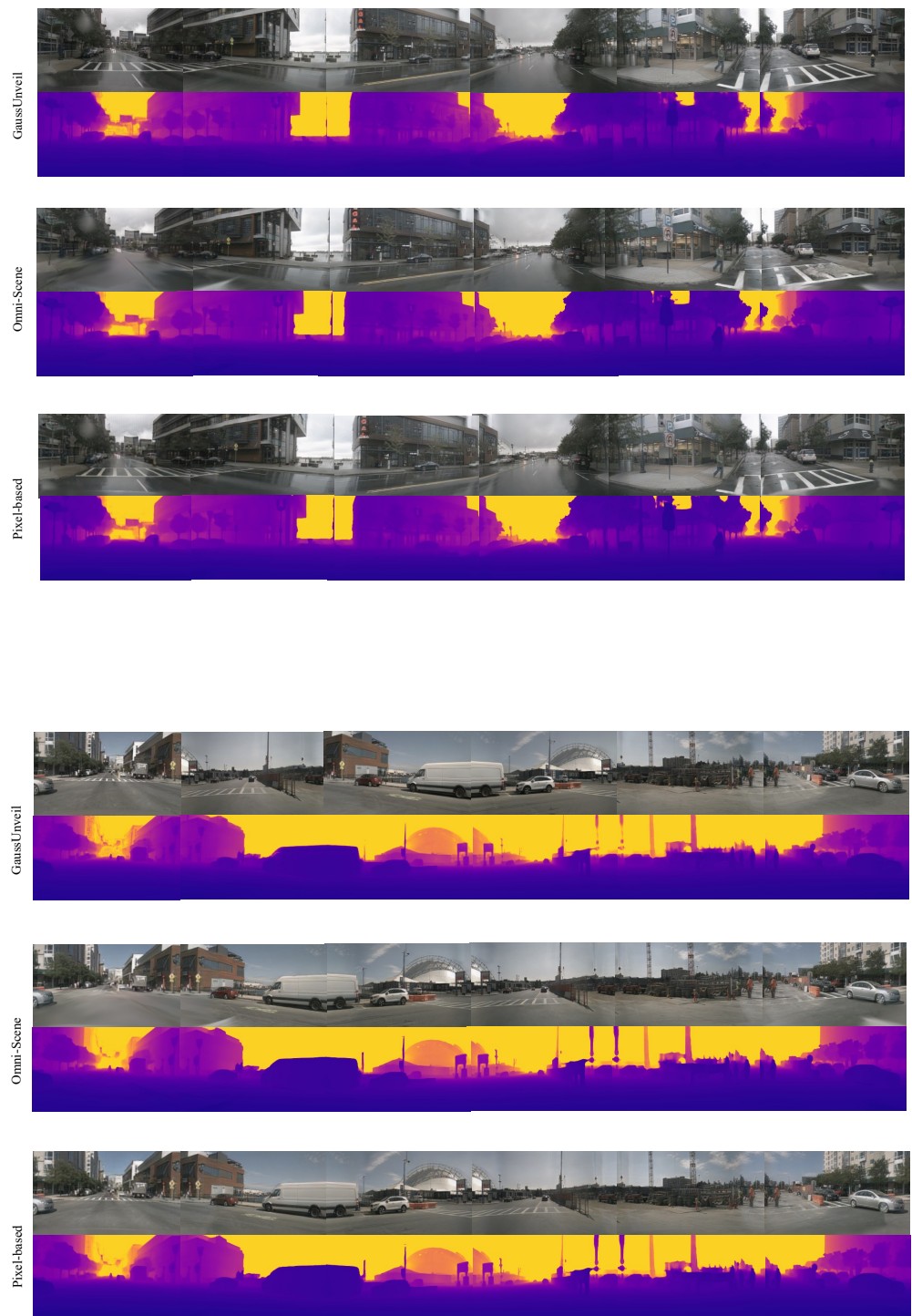

Figure 5: The qualitative comparison of reconstruction performance between Omni-Scene (Wei et al., 2025), pixel-based method, and our GaussUnveil (better viewed when zoomed in). We render six views to cover the full 360° panorama, ensuring approximately 15% overlap between adjacent viewpoints. The red boxes indicate the overlapping regions across different views.

