# OpenReview forum: "GaussUnveil: Unified Occlusion-Aware Gaussian Refinement for Sparse-View Scene Reconstruction"
_ICLR.cc/2026/Conference — ICLR 2026 Conference Withdrawn Submission_

### Official Review · Reviewer_ADsJ · 2025-10-28

**Soundness:** 3
**Presentation:** 2
**Contribution:** 2
**Rating:** 2
**Confidence:** 4

**Summary:**

The paper proposes GaussUnveil, a unified Gaussian-based framework for sparse-view 3D scene reconstruction. Unlike prior dual-branch methods such as OmniScene, which refine all regions uniformly, GaussUnveil selectively refines Gaussians only in occlusion-prone regions identified through depth-gradient–based occlusion masks. The occluded regions are lifted into 3D visibility transition tubes, where a lightweight 3D Gaussian refinement block updates the mean and covariance of the local Gaussians via self-context and cross-view aggregation. The paper show that GaussUnveil reduces Gaussian count faster inference while maintaining reconstruction qualtiy compared to existing works.

**Strengths:**

- Clear motivation and intuitive core idea on refining only occlusion-prone regions.
- Strong presentation and clarity in which the key concepts are well explained and illustrated.
- Detailed appendix provides sound theoretical grounding for the refinement paradigm.

**Weaknesses:**

1. **Marginal quantitative improvement** — gains are small (<0.5 dB PSNR) and inconsistent across datasets; improvements are more visible on nuScenes than on RealEstate10K.
2. **Limited novelty** — the approach mostly repurposes known ideas (depth-edge detection + Gaussian refinement) rather than introducing a fundamentally new representation.
3. **Efficiency claim lacks detailed evidence** — parameter counts are nearly identical to OmniScene (<2% difference), yet runtime is reported 30% faster without profiling or explanation.
4. **Missing visualization of refinement effects** — the paper would benefit from showing how individual Gaussians are refined or pruned during training to support its key claim.
5. Minor figure inconsistency — red boxes mentioned in Fig. 5 caption are missing.

**Questions:**

- It would be great if the authors **quantify the source of the runtime speedup** compared to OmniScene.
- How sensitive is performance to the **occlusion-mask hyperparameters** (τg , k , κrel, κabs)? Are these tuned per dataset?
- During **occlusion mask lifting**, is the **number of sampled Gaussians** a fixed hyperparameter or adaptive?
- Could the authors include **visualization of refinement**—before/after Gaussian distributions or occlusion-region overlays—to make the selective refinement effect tangible?

---

### Official Review · Reviewer_dENn · 2025-10-29

**Soundness:** 2
**Presentation:** 2
**Contribution:** 1
**Rating:** 2
**Confidence:** 3

**Summary:**

This paper introduces a feed-forward Gaussian scene reconstruction model that selectively refines potential occlusion-prone regions instead of uniformly refining all Gaussians.
The proposed method GaussUnveil uses depth-gradient masks to identify occlusion uncertain areas (near visibility transition) and applies a lightweight refinement block to update local Gaussian parameters. Experiments on ego-centric and scene-centric benchmarks show that GaussUnveil achieves comparable or better reconstruction quality with a fewer number of Gaussians and faster inference than existing methods such as Omni-Scene.

**Strengths:**

Overall, I found the paper to have a clear motivation, addressing the efficiency and occlusion-robustness challenges in feed-forward Gaussian reconstruction.

The proposed occlusion-aware selective refinement mechanism is conceptually simple yet practical, leading to reduced Gaussian count and faster inference without sacrificing reconstruction quality.

This paper achieves state-of-the-art performance on several benchmarks, showing the effectiveness of this proposed approach and architecture.

**Weaknesses:**

I think the paper presents a well-engineered framework, but the conceptual novelty appears limited.

**W1.** The proposed selective-refinement mechanism mainly reorganizes existing feed-forward Gaussian pipelines such as MVSplat or Omni-Scene, focusing on occlusion-prone regions without introducing a fundamentally new representation or learning paradigm.

**W2.** The claimed robustness under sparse or low-overlap settings is not convincingly demonstrated; the visibility-transition mask relies on depth-gradient estimation, which itself presumes reliable multi-view geometry—contradicting the sparse-view motivation.

**W3.** Most architectural components and training objectives are inherited from the baseline methods, making the improvement primarily an efficiency-oriented variant rather than a conceptually novel framework.

**Questions:**

**Q1.** How does the method quantitatively perform under varying degrees of view overlap?
(e.g., does selective refinement still help when overlap becomes extremely low?)

**Q2.** How robust is the visibility-transition mask when the depth estimation is noisy or inaccurate? I understand that the depth prediction might not be perfect, but the architectural design might still compensate for errors through feature aggregation — could the authors clarify whether such robustness is empirically validated or explicitly modeled during training?

**Q3.** What is the key component that makes your pipeline much faster than pure pixel-based methods? I understand that the pipeline includes additional steps such as depth prediction and refinement, which would normally increase computation. Could you clarify which design choice (e.g., reduced Gaussian count, parallelized refinement, or omission of voxel fusion) primarily contributes to the observed runtime improvement?

---

### Official Review · Reviewer_zYJs · 2025-10-30

**Soundness:** 3
**Presentation:** 3
**Contribution:** 3
**Rating:** 6
**Confidence:** 4

**Summary:**

In this paper, the authors propose a framework named GaussUnveil, which predicts pixel-aware Gaussians and performs 3D refinement on occlusion-prone regions to improve the accuracy in occluded areas. Specifically, GaussUnveil localizes unreliable geometry in occlusion-prone regions using sharp depth-gradient boundaries. Then it uses the Refine Block to update the 3D attributes of these Gaussians to correct geometric deviations caused by occlusion. The experiments show that GaussUnveil achieves SOTA results.

**Strengths:**

1. The authors focus on the trade-off between the number of Gaussians and rendering performance: increasing the number of Gaussians typically improves rendering results, but it also raises memory consumption. To address this memory issue in Gaussian rendering, the authors selectively refine only occlusion-prone regions—ultimately achieving comparable rendering quality with far fewer Gaussians.
2. The 3D Gaussian Refinement module effectively improves rendering results.
3. What's more, they demonstrate the superiority of this framew

**Weaknesses:**

1. This method is based on pixel-aware Gaussian modeling and relies heavily on high-quality depth initialization. Thus, poor depth initialization may propagate errors to subsequent 3D refinement steps and ultimately affect the overall reconstruction accuracy.
2. The paper lacks sufficient discussion on hyperparameters critical to the method’s performance, such as those used to determine the boundary.

**Questions:**

If the authors discuss whether GaussUnveil achieves good performance in cross-dataset generalization, like how MVSplat uses a model trained on Re10k and then evaluates it on DTU or other datasets, this would enhance the method’s practical validation.

---

### Official Review · Reviewer_Coke · 2025-10-31

**Soundness:** 3
**Presentation:** 2
**Contribution:** 2
**Rating:** 4
**Confidence:** 4

**Summary:**

This paper proposes GaussUnveil, an occlusion-aware 3D Gaussian refinement framework for sparse-view scene reconstruction, which selectively refines only regions near occlusions instead of refining all Gaussians uniformly.
Introduces a visibility transition localization module that generates occlusion masks from depth-gradient discontinuities and lifts them into 3D to guide refinement.
Designs a lightweight 3D Gaussian Refinement Block with self-context aggregation, cross-view aggregation, and corrective refinement layers to update Gaussians in masked regions.
Evaluates on nuScenes (ego-centric) and RealEstate10K (scene-centric), achieving state-of-the-art results: on nuScenes, 24.65 PSNR, 0.754 SSIM, 0.220 LPIPS, 0.837 PCC, with 30% fewer Gaussians and 34% faster than Omni-Scene.

**Strengths:**

GaussUnveil presents a well-motivated and practical approach to sparse-view 3D reconstruction by introducing an occlusion-aware selective refinement strategy that significantly reduces redundancy in 3D Gaussian representations. Its originality lies in shifting from uniform refinement to “refine where it matters,” using depth-gradient-based masks to localize visibility transitions—a simple yet effective insight. The technical quality is strong, with thorough experiments on both ego-centric (nuScenes) and scene-centric (RealEstate10K) benchmarks, clear ablations, and efficiency gains (30% fewer Gaussians, 34% faster). The paper is clearly written with intuitive figures, and its significance is notable: it addresses a real bottleneck in Gaussian-based reconstruction for autonomous driving and generalizes across settings, offering a scalable paradigm for efficient 3D scene modeling.

**Weaknesses:**

1 The paper lacks sufficient visualization and analysis of the occlusion masks: it is unclear how reliably the depth-gradient-based masks capture true occlusion regions, especially for viewpoint-induced occlusions that may not produce sharp depth edges—more qualitative examples and failure cases are needed.

2 Qualitative comparisons are inconclusive: all rendered novel views in Figure 4 and Figure 5 appear nearly identical across methods, making it hard to discern improvements; the paper should include more challenging or extreme viewpoints (e.g., highly oblique or occluded angles) to better validate reconstruction in previously hidden regions.

3 On RealEstate10K, GaussUnveil shows only marginal gains over MVSplat and underperforms in PCC compared to some baselines; the paper does not compare against recent sparse-view SOTA methods like TranSplat or DepthSplat, leaving its actual advantage in scene-centric settings unclear.

**Questions:**

Similar to Weaknesses.

---

### Note · Authors · 2025-11-12

I have read and agree with the venue's withdrawal policy on behalf of myself and my co-authors.